# Can immersive virtual reality magnify treatment outcomes of computerized script training on Cantonese speakers with chronic aphasia? Protocol of a randomized controlled trial

**Winsy Wing Sze Wong**[1]*, **Donald Shi Pui Li**[2], **Kenneth Ngai Kuen Fong**[3],
**Peter Hiu Fung Ng**[4], **Hoi Tsz Karen Kwok**[1]

1 Department of Language Science and Technology, Faculty of Humanities, The Hong Kong Polytechnic University, Hung Hom, Kowloon, Hong Kong SAR, China, 2 Department of Cognitive Science, Krieger School of Arts and Sciences, Johns Hopkins University, Baltimore, Maryland, United States of America, 3 Department of Rehabilitation Sciences, Faculty of Health and Social Sciences, The Hong Kong Polytechnic University, Hung Hom, Kowloon, Hong Kong SAR, China, 4 Department of Computing, Faculty of Engineering, The Hong Kong Polytechnic University, Hung Hom, Kowloon, Hong Kong SAR, China

* winsyws.wong@polyu.edu.hk

## Abstract

### Background

Aphasia persists in 20% of the stroke population. Speech therapy has been suggested to benefit the functional communication of people with aphasia (PWA). Virtual reality (VR) has been widely applied in motor/cognitive rehabilitation of stroke patients. The current proposal investigates the efficacy of VR-based computerized script training on the functional communication of PWA.

### Methods

A three-armed assessor-blinded randomized controlled trial (RCT) will recruit 120 PWA. They will be randomly assigned to 1) VR-based computerized script training, 2) computerized script training without VR, or 3) no treatment control. In the VR condition, PWA will practice in VR depicting everyday contexts, while in the non-VR group, PWA will practice with static photos. PWA in the control group will be assessed without training. Outcomes on functional communication, aphasia severity, and quality of life will be compared before/after treatment. The findings will provide empirical evidence to inform the development of effective and ecologically valid interventions for PWA.

### Trial registration

ClinicalTrials.gov NCT06722092

**Data availability statement:** No datasets were generated or analysed during the current study. All relevant data from this study will be made available upon study completion.

**Funding:** Wong received the General Research Fund from Hong Kong Research Grants Council for the project (Project number: 15602024). https://www.ugc.edu.hk/eng/rgc/funding_opport/grf/ The funders had no role in study design, data collection and analysis, decision to publish, or preparation of the manuscript.

**Competing interests:** NO authors have competing interests.

## Introduction

Aphasia, a deficit in language comprehension and production predominantly caused by a stroke, affects about 20% of post-stroke populations in the chronic stage [1]. It imposes a significant impact on the communication of people with aphasia (PWA) in different social/vocational contexts, affecting their well-being [2]. A Cochrane review [3] concluded that speech-language therapy (SLT), in contrast with social support or communication stimulation that does not target specific therapeutic goals, is effective in reducing aphasia severity and improving functional communication (i.e., language use in interactive, multimodal, and situated contexts [4]). Different interventions, with targets at varying levels of linguistic complexity, delivered on a group/individual basis, conducted by real-person clinicians/computers, have been developed and shown to be beneficial for the communication and language processing of PWA. However, treatment methods vary in effect size, and there is limited understanding of the 'active ingredients' within these protocols that play critical roles in enhancing treatment outcomes. Factors, including dose (see a review by Harvey et al. [5]), treatment intensity [6], and cueing conditions [7], that affect treatment effectiveness/efficiency have been investigated. Alternatively, another research direction concerns the application of cutting-edge technologies in treatment delivery, e.g., mobile apps and virtual reality (VR), and their potential to further enhance treatment outcomes compared with conventional SLT.

VR, defined as the use of computer technology to create a realistic environment, enables users to immerse themselves, interact, and experience a sense of presence in the simulated environment [8]. The application of VR in rehabilitation is more than a novel and enjoyable experience; its features and the underlying reasons for promoting learning/treatment outcomes are supported by neurolinguistic theories. It is argued that the immersive experience and multi-sensory feedback provided by VR allow users to perform motor training and observe the actions displayed by avatars, which would subsequently activate the mirror neuron system [9] and promote recovery [10]. Another neurolinguistic account, known as the theory of embodied cognition, explains that the multimodal interactions between the user and the virtual environment provided by VR in a social context may facilitate conceptual processing and its representation (see Mahon & Caramazza [11] for a discussion). In addition, there is both neural and behavioral evidence that the sensory experience in VR can change neural representations and behavioral decisions outside VR when neurologically healthy participants encounter similar situations in the future [12]. Although the theoretical account of VR in language rehabilitation for post-stroke aphasia remains inconclusive, how VR benefits second-language learning from an embodied and social cognition perspective has been discussed based on behavioral and neuroimaging findings [13]. Given that the sense of embodiment and presence in VR is also readily experienced by people suffering from stroke [14], it is, therefore, reasonable to postulate that stroke patients may likewise benefit from the immersive and interactive training context provided by VR, with a possibility of transferring the skills learnt from therapy to everyday life contexts.

The application of VR in aphasia therapy is relatively new. A systematic review and meta-analysis [15] based on five RCTs/quasi-RCTs concluded that VR-based

interventions had a marginally positive clinical effect on aphasia severity compared to conventional treatment, but showed no significant impact on functional communication, repetition, or naming. However, only three of the studies [16–18] adopted a functional approach to treatment, and outcomes for functional communication were reported in Marshall et al. [16] and Grechuta et al. [17]. Some studies [17,19] employed different treatment protocols and content across VR and non-VR therapy, making it difficult to isolate VR's specific contribution. Besides, semi-immersive VR was used in most studies except in Zhang et al. [19], where immersive VR was applied. Despite the mixed findings, a more recent review [20] suggested that VR offers a unique opportunity to address rehabilitation at the impairment, activity, and participation levels. We also argue that a more rigorous methodology, i.e., an RCT that uses the same treatment content delivered in VR vs. a non-VR-based intervention, would allow us to compare the crucial features of VR, i.e., immersion and interactivity, in aphasia intervention.

## Aims and hypotheses to be tested

The proposed study aims to examine whether VR-based computerized script training may improve verbal functional communication in PWA and, more importantly, whether such therapy may enhance aphasia treatment outcomes. Immersive VR with 360-degree panoramic scenes, a lower-cost option than three-dimensional modeled applications yet effective at creating a sense of presence [21], will be delivered via a head-mounted display (HMD). The use of immersive VR is supported by evidence suggesting that greater immersion may better promote treatment outcomes in stroke patients receiving motor rehabilitation (see a review and meta-analysis by Palacious-Navarro & Hogan [22]) and in social skills learning in some individuals with autism spectrum disorder [23]. Script training, a functional approach to aphasia rehabilitation [24], will be chosen due to several reasons. Firstly, it is an evidence-based intervention (see Hubbard et al. [25] for a review) for fluent and non-fluent PWA. Monologic or dialogic scripts are functional to the daily needs of PWA, e.g., ordering dishes at a restaurant or making a doctor's appointment, can be prepared and practiced following a cueing hierarchy predefined in the computer program to avoid clinician-led variability in style and thus ensure higher adherence to the treatment protocol. Secondly, the immersion and interactivity provided by VR align perfectly with the theoretical underpinnings and the context-bound nature of script training. The instance theory of automatization [26] explains that script acquisition (i.e., automatic retrieval of script content) is achieved via retrieving memories of complete, context-bound, and skilled performance through repeated exposure to and practice on the same task [27]. The immersive experience delivered via an HMD, together with interactions between the PWA and the virtual conversation partner in the interactive videos, will help create a naturalistic and complete context for the PWA to practice the script and eventually achieve automatic retrieval in everyday communication. Last but not least, VR-based therapy allows PWA who may have limited access to multiple locations due to physical constraints and reduced confidence in communication post-stroke to practice the scripts in a safe, repetitive, but less embarrassing manner. Such an unparalleled experience, when compared to the conventional clinical setting, may further enhance the treatment outcomes of script training.

## Research questions of the current proposed study

(1) Can VR-based computerized script training promote verbal functional communication of Cantonese-speaking PWA? This question will be addressed by comparing participants' performance in the VR treatment condition with that in the no-treatment control across different timepoints of the study. It is hypothesized that the verbal functional communication of PWA in the VR treatment group will improve, while those in the control group will not show significant changes.

(2) Is VR-based computerized script training more efficacious than computerized script training without VR in promoting verbal functional communication of Cantonese-speaking PWA? This question will be addressed by comparing participants' performance in the VR treatment condition with that in the non-VR condition.

## Materials and methods

### Pilot data

A pilot study (reference number of the Human Subjects Ethics Application Review System of the Hong Kong Polytechnic University Institutional Review Board: HSEARS20221021003–01) has been conducted to: 1) develop the computerized script training for Cantonese PWA, 2) develop the materials and program for script training via VR, and 3) demonstrate feasibility and preliminary efficacy of computerized script training with/without VR. Based on the pre/post comparison of six PWA who were randomly assigned to receive either VR/non-VR treatment from June to August 2023, large effect sizes were obtained in the primary outcome measure (in terms of percentage of words correctly produced without written cues) in both VR (Cohen's $d = 2.03$) and non-VR treatment groups (Cohen's $d = 5.52$). For secondary outcomes, generalization was observed in both groups, with Cohen's $d = 1.84$ and $6.95$ for VR and non-VR groups, respectively. Verbal functional communication, as measured by the Cantonese version of the Amsterdam Nijmegen Everyday Language Test (CANELT) [28], had demonstrated some improvements in the VR (Cohen's $d = 0.75$) and the non-VR group (Cohen's $d = 0.36$). Positive changes in the aphasia quotient were observed, with a Cohen's $d$ of $1.36$ and $0.53$ in the VR and non-VR groups, respectively. Adherence to treatment protocol was similar in both treatment groups. PWA reported a satisfactory experience with computerized script training and VR exposure, with minimal cybersickness, as reflected by an average comfort level rating of 4.6/5 in the VR group. Despite the seemingly unequal aphasia severity in both groups before treatment (mean AQ of 56.4 and 79.0 in VR and non-VR groups, respectively), clinically meaningful changes were observed in both treatment groups in most of the language and communication measures, while some of the secondary outcome measures in the VR group had larger effect sizes than the non-VR group. In sum, the preliminary results have demonstrated the feasibility of both computerized script training and VR-based script practice proposed by the research team, providing strong support for pursuing the current proposal.

### Study design

A prospective, three-armed, open-label, assessor-blinded, randomized controlled trial will be conducted over 20 weeks. Protocol design and data reporting are aligned with the Consolidated Standards of Reporting Trials Statement (CONSORT) and Standard Protocol Items: Recommendations for Interventional Trials (SPIRIT) for nonpharmacologic trials [29]. The treatment protocol (version 1.0, dated July 24, 2024) has been registered in ClinicalTrials.gov (NCT06722092). The experimental design is summarized in the S1 Fig.

### Ethics statement

Ethical approval has been obtained from the Human Subjects Ethics Application Review System of the Hong Kong Polytechnic University Institutional Review Board (reference number: HSEARS2023022800302). Written consent will be sought from the eligible participants in the presence of caregivers.

### Participants

A total of 120 Cantonese-speaking PWA will be recruited from rehabilitation centers, stroke patient support groups in Hong Kong, and speech therapy clinics at a local institute. The sample size estimate is conducted using G* power (version 3.1.9.7) repeated-measure, within-between interaction ANOVA, based on 3 groups, 5 repetitions, power of 95% with $\alpha = .003$ (adjusted for a total 16 multiple comparisons: 3 between groups (VR vs. control, non-VR vs. control and VR vs. non-VR), 3 between timepoints (pre vs. post, pre vs. mid and post vs. mid) and 10 interaction between groups and timepoints), a medium primary outcome effect size of Cohen's $d = 0.5$ ($\eta_p^2 = 0.0588$), a 0.5 correlation among repetitive measures, a 0.5 non-sphericity correction and an attrition rate of 20%. A projected 20% attrition rate was incorporated into the power analysis to account for potential loss to follow-up. This estimate is informed by historical data from similar protocols,

in which dropout rates of 10–20% occurred due to health fluctuations, logistical barriers (e.g., transit difficulties), or waning participant engagement. Attrition is assumed to be non-differential across study arms, as the anticipated reasons for withdrawal are independent of the specific intervention. We assumed a medium effect size of 0.5 based on the smallest effect size of $d = 0.74$ obtained from Cherney et al. [7], a study on computerized script training with a comparable total treatment duration.

Inclusion criteria include: 1) a stroke onset of more than six months, with an Aphasia Quotient (AQ) below 96.4, as evaluated by the Cantonese version of Western Aphasia Battery (CAB) [30], 2) premorbid fluent Cantonese speakers, 3) aged between 30 and 80 years, 4) no reported progressive neurogenic disorders such as dementia or Parkinson's disease, 5) no motor speech disorders of moderate to severe level, based on a rating of 3 or above in the Therapy Outcome Measure [31,32], 6) normal or corrected-to-normal vision and hearing functions. Exclusion criteria include: 1) presence of neurological disorders, e.g., dementia or motor neuron disease; and 2) non-compliance with a 10-minute VR exposure during screening, such as reports of cybersickness or discomfort. Recruitment through the collaborating rehabilitation centers and stroke patient support groups will be done via posters and social media posts. The screening will be done by a trained research postgraduate student/research assistant with a speech-language pathology background.

### Randomization

Co-author DL, who is not involved in participant recruitment, condition assignment, and assessment/ treatment throughout the study, will prepare a central web-based randomization sequence, with an allocation ratio of 1:1:1 to the 1) VR-based treatment condition, 2) non-VR treatment condition, and 3) no-treatment condition. Participant enrolment will be conducted by co-author HTK, and condition assignment will be conducted by author WW and a trained research assistant who is not involved in the enrolment, screening, or assessment of PWA. Simple randomization will be used. PWA who have given informed consent will be randomly allocated to one of the above-mentioned conditions. All participants in the no-treatment group will receive VR/non-VR-based treatment upon the completion of all assessments.

### Assessment and treatment schedule

Participants will receive pre-treatment assessments twice, i.e., at three and one weeks pre-treatment, as baseline. The same set of assessment tasks will be administered mid-treatment, within a week post-treatment, and 8 weeks post-treatment as maintenance. The treatment schedule will be the same for both treatment conditions: 14 therapy sessions will occur twice per week from week 5 to week 7 and from week 9 to week 11.

### Outcome measures and procedures

Primary outcomes: During an assessment session, the PWA's accuracy and time needed to produce the trained scripts (i.e., treatment probe) in read-aloud and no-cue conditions will be recorded via the computerized script training software. Similar to the training session, the virtual therapist will initiate the conversation, followed by PWA's response. The no-cue condition will be executed first, followed by the read-aloud condition. The number of sessions required to reach the performance criterion (i.e., 90% accuracy in two consecutive sessions) will also be considered to reflect PWA's progress.

Secondary outcomes: To evaluate treatment generalization, the assessment session will include novel scripts that were not trained but share similar contexts with the trained scripts. Examples include ordering dishes that differ from the trained items or eliciting untrained personal information during a self-introduction. In accordance with the Research Outcome Measurement in Aphasia (ROMA) [33], the aphasia quotient of CAB [30], is included as an outcome measure instrument (OMI) for language ability. Since a Hong Kong-adapted version of The Scenario Test [34] is not available, CANELT [28], a standardized OMI adapted from the Amsterdam-Nijmegen Everyday Language Test (ANELT) [35] for Cantonese speakers in Hong Kong, will be included as secondary outcomes in language and communication. CAB [30] is a standardized aphasia test for measuring aphasia severity. CANELT [28] assesses the effectiveness of verbal communication in conversation

through 20 culturally appropriate scenarios. In addition, PWA's self-reported OMI on communication and quality of life will be examined via the Cantonese version of the 20-item Communication Outcome after Stroke (Can-COAST) Scale [36] translated from its original English version (COAST) [37]. All assessment tasks and their sources of normative data (if applicable) are summarized in Table 1.

Assessment will be done by a trained research postgraduate student with healthcare background, such as speech therapy or rehabilitation science, who is blinded from the therapy condition of subjects. S/he will not be involved in the intervention process. A standardized procedure with audio recording for outcome assessment will be enforced. Primary outcomes and generalization probes are computerized and audio-recorded to minimize and monitor any assessor's bias. Since blinding participants may be deemed impossible, both the assessor and the PWA are reminded not to mention or discuss the therapy condition assignment during assessment sessions.

## Treatment materials and procedures

A total of 14 individual treatment sessions, given twice/three times per week, will be delivered to both treatment groups. Each session will include a 30-minute computerized script training, followed by a 30-minute practical section delivered with or without VR. No intervention will be given to the no-treatment control group during the study period. However, PWA in the control group will receive either VR or non-VR-based script training for free upon study completion. All treatment sessions will be done in a quiet room in the speech therapy unit by a trained research postgraduate student/research assistant with a speech-language pathology, linguistics, or rehabilitation science background. Since the treatment providers cannot be blinded to treatment condition assessment, the following procedures will be adopted to mitigate their potential expectancy effects: 1) treatment providers will receive standardized training prior to therapy delivery, based on a standardized protocol and therapy manual developed by the authors (WW and HTK). Scripted instructions and feedback will be given. Competency evaluation will be conducted through role-play before intervention delivery; 2) all treatment materials and procedures are standardized and computerized. All training logs and verbal productions during therapy sessions are recorded digitally for monitoring. Treatment providers will fill out a log sheet for each session for recording incidents when protocol is not followed, which will be submitted the principal investigator of the project within the same day, 3) the performance of the treatment providers will be monitored by author (WW) and an independent speech therapist

**Table 1. List of assessment and outcome measures and sources of normative data.**

| Primary outcome | Sources of normative data |
|---|---|
| Percentage accuracy of treatment probe correctly produced in no-cue and read-aloud conditions | NA |
| Time to produce the scripts in the treatment probe in no-cue and read-aloud conditions | NA |
| Number of sessions to reach performance criteria in session probe | NA |
| Secondary outcomes | |
| Aphasia quotient of CAB | Yiu (1992) [30] |
| Age-adjusted z scores of CANELT | Wong (2024) [28] |
| Total scores of Can-COAST | Wong and Kwok (2025) [36] |
| Generalization | |
| Percentage accuracy of generalization probe correctly produced | NA |
| Time to produce the scripts in the generalization probe | NA |

CAB, Cantonese version of the Western Aphasia Battery; CANELT, Cantonese version of the Amsterdam-Nijmegen Everyday Language Test; Can-COAST, Cantonese version of the Communication Outcome after Stroke Scale. NA, Not applicable.

not involved in the study for fidelity checking (see Section 'Reliability and treatment fidelity'). The principal investigator and the project team will discuss corrective actions for protocol deviations.

## Script development and selection

Six to eight dialogic scripts were developed via pilot study, each script containing five conversational turns, will be used for training (i.e., treatment probe). Each PWA will be interviewed to identify the scripts most relevant and functional for their daily needs. The number of syllables and grammatical complexity in the scripts will be balanced across participants.

## Computerized script training

A computerized script training program delivered in Cantonese will be used with procedures adapted from Cherney et al. [27]. The first author has already developed the program supported by an internal grant provided by the institute. Each treatment session begins with a session probe, which monitors the PWA's performance on scripts trained in previous sessions without support/cueing. If the performance on a certain script exceeds 90% in two consecutive sessions, a new script will be introduced. Treatment follows the sequence of 1) The whole conversation script will be presented on a computer screen and read aloud by the virtual clinician, 2) The PWA will read each conversational turn assigned twice in unison with the virtual therapist. 3) The PWA will read aloud the sentence independently while the program will record the verbal response, 4) Treatment proceeds to another conversational turn. Each script will be repeated twice while two to three scripts will be trained in each session in random order. A clinical assistant will be present to monitor and provide technical support. Following the computerized script training, the same sets of scripts will subsequently be practiced for 30 minutes, either in the VR or non-VR condition. The same set of procedures will be applied in the two treatment conditions; they only differ in terms of how treatment materials are presented.

## VR-based script training

The PWA will wear an HMD device for 30 minutes. Interactive 360-degree videos depicting various daily scenarios will be presented. The immersive experience provided by the 360-degree videos will include a visual exploration of the scenario/surroundings (e.g., walking around in a fashion shop/reading a food menu in a restaurant) and audio input via stereo speakers provided by the HMD (traffic on the street, background noise of a Chinese restaurant, etc.). Interactive components will be achieved mainly via communication with the virtual communication partner and its induced events. For example, a waiter in the restaurant will initiate a conversation. The PWA will be given time to respond. If a correct response is produced, the virtual communication partner will 'interact' with the PWA by initiating the next conversational turn, and, if applicable, nonverbal reactions/rewards such as presenting the object requested to the PWA. For incomplete or erroneous productions, PWA will be required to repeat the correct response in unison with the conversational partner. The clinical assistant, who accompanies the PWA in the entire training session, will decide and control the type of feedback. Rests within the session will be given, and any possible adverse effects will be monitored and reported.

## Script training without VR

The training flow in the non-VR condition is identical to the VR condition, except that only static photos depicting the scenarios and pre-recorded conversational turns will be presented to the PWA on a computer screen.

## Data analysis

Assessment will be administered by trained research personnel with speech-language pathology background, and therapy will be delivered by trained research assistants or speech-language pathology students who are blinded to the assignment to the treatment condition. Transcriptions will be performed by trained research assistants with backgrounds in linguistics

or speech-language pathology. Scoring will be done by the fifth author (HTK), trained speech-language pathology students, and trained personnel with a speech-language pathology or linguistics background. Protocols and guidelines for transcription and scoring will be regularly reviewed by the first (WW) and fifth (HTK) authors. Data analyses will be carried out by the first (WW) and fifth (HTK) authors. Treatment and generalization probes will be scored by i) the percentage of target syllables correctly produced and ii) the Naming and Oral Reading for Language in Aphasia 6-point scale (NORLA-6) with some adaptations. NORLA-6 has been validated [38] and adopted in a number of script training studies [7,39]. Syllable accuracy will be measured by strict alignment with the personalized script. Partial utterances, as well as semantic and phonemic paraphasia, will be evaluated under the NORLA-6 scale criteria.. Table 2 shows the adapted scoring criteria of NORLA-6. Each word produced by the participant will be given a score from zero to five based on fluency, accuracy, and the nature of errors produced. Adaptations were made to the scoring criteria for Scores 3 and 4; specifically, two conditions will be removed as Cantonese lacks inflectional verb tense and prosodic variation (e.g., stress on incorrect syllable). The remaining scoring criteria will adhere to the NORLA-6 framework to ensure consistency. For instance, the best production will be scored when multiple attempts are made. Commentary productions that are not intended for the target word will not be scored. The efficiency of script production in terms of words produced per minute will also be calculated. The timing window will begin at the completion of the prompt to the participant's vocal offset.

For secondary outcomes, PWA's performance on CANELT [28] will be transformed into z-scores based on the normative dataset collected from 100 healthy Cantonese speakers aged between 30 and 79 years. Meanwhile, raw scores of CAB [30] and Can-COAST [36] will be used. Descriptive statistics will first be computed for all outcome measures and PWA characteristics. To examine whether baseline performance is stable across the two pre-treatment assessments, we will use repeated-measure ANOVA to test whether the scores at the two pre-treatment timepoints are comparable across all outcomes. Outcomes with comparable baseline performance will be averaged across the two timepoints for subsequent analyses; conversely, the timepoint with better performance will be treated as baseline for subsequent analyses. Separate linear mixed effect (LME) models with identity link function will be used to investigate whether the use of VR on script training (i.e., experimental conditions) enhances each of the primary and secondary outcomes across timepoint. In each model, experimental conditions, timepoints (categorical variable with three levels – pre-, mid- and post-treatment – to allow for the detection of non-linear treatment effects and to facilitate pairwise comparisons between specific assessment

**Table 2. Adapted naming and oral reading for language in Aphasia 6-Point Scale (NORLA-6).**

| 5 | Accurate and immediate response |
|---|---|
| 4 | Accurate but delayed response (A delay is defined as: a pause, with or without fillers, preceding the target word that is atypical of the participant's oral reading and naming performance. Some participants may show consistent, delayed responses in the context of slow and effortful oral reading and naming. In such cases, all words are rated a 4.)<br>Self-corrected response<br>Minor distortion of a recognizable phoneme<br>~~Prosodic variation (e.g., stress on incorrect syllable)~~<br>Correct word produced in the wrong order in the sentence |
| 3 | Omission, addition, or substitution of a grammatical morpheme<br>Omission, addition, or substitution of a single phoneme (regardless of the length of the word)<br>~~Incorrect use of verb tense for irregular verbs~~ |
| 2 | Related verbal paraphasia (includes in-class substitutions of function words such as determiners, prepositions, pronouns)<br>Verbal paraphasia with a minor phonemic error<br>Phonemic paraphasia in which more than half of the word is correct |
| 1 | Unrelated verbal paraphasia as a substitution for the target word<br>Unrelated response as a substitution for the target word<br>Unintelligible response as a substitution for the target word<br>Phonemic paraphasia in which half or less of the word is correct |
| 0 | No verbal response |

sessions) and their interactions will be fixed factor and one primary/secondary outcome will be used as the dependent variables. Covariates that may interfere with the outcomes, including age, aphasia severity, baseline performance, and education, will be included as fixed factors in the LME models, while random intercept for participant and stimulus item (if appropriate) will also be included in the LME models. In sum, the following statistical models will be used to investigate the treatment outcomes:

Primary outcomes:

(1) Percentage accuracy of treatment probe in the no-cue condition $\sim 1 + $ timepoint*condition $+$ baseline performance $+$ age $+$ education $+$ baseline CAB score $+ (1\,|\,$subject$) + (1\,|\,$stimulus item$)$

(2) Percentage accuracy of treatment probe in the read-aloud condition $\sim 1 + $ timepoint*condition $+$ baseline performance $+$ age $+$ education $+$ baseline CAB score $+ (1\,|\,$subject$) + (1\,|\,$stimulus item$)$

(3) Total time to produce the scripts in the treatment probe in the no-cue condition $\sim 1 + $ timepoint*condition $+$ baseline performance $+$ age $+$ education $+$ baseline CAB score $+ (1\,|\,$subject$) + (1\,|\,$stimulus item$)$

(4) Total time to produce the scripts in the treatment probe in the read-aloud condition $\sim 1 + $ timepoint*condition $+$ baseline performance $+$ age $+$ education $+$ baseline CAB score $+ (1\,|\,$subject$) + (1\,|\,$stimulus item$)$

(5) Number of sessions required to reach the performance criterion in probing session $\sim 1 + $ timepoint*condition $+$ baseline performance $+$ age $+$ education $+$ baseline CAB score $+ (1\,|\,$subject$) + (1\,|\,$stimulus item$)$

Secondary outcomes:

(6) CAB $\sim 1 + $ timepoint*condition $+$ baseline performance $+$ age $+$ education $+ (1\,|\,$subject$)$

(7) CANELT $\sim 1 + $ timepoint*condition $+$ baseline performance $+$ age $+$ education $+$ baseline CAB score $+ (1\,|\,$subject$)$

(8) Can-COAST $\sim 1 + $ timepoint*condition $+$ baseline performance $+$ age $+$ education $+$ baseline CAB score $+ (1\,|\,$subject$)$

**Generalization.**

(9) Percentage accuracy of generalization probe $\sim 1 + $ timepoint*condition $+$ baseline performance $+$ age $+$ education $+$ baseline CAB score $+ (1\,|\,$subject$) + (1\,|\,$stimulus item$)$

(10) Total time to produce the generalization scripts $\sim 1 + $ timepoint*condition $+$ baseline performance $+$ age $+$ education $+$ baseline CAB score $+ (1\,|\,$subject$) + (1\,|\,$stimulus item$)$

The variance inflation factor (VIF) values in the mixed models will be used to examine multicollinearity of the predictors. If the VIF values are large ($\geq 10$), those highly correlated predictors will be combined into a composite before entering the mixed models. Subjects with incomplete longitudinal data will be retained in the LME model analysis as the LME model estimation using maximum likelihood handles missingness at random well. Any significant results obtained in the above-mentioned LME models will be followed by post-hoc comparisons with Bonferroni correction. Such planned subgroup analyses will be exploratory, aiming to investigate outcomes across different conditions. Model assumptions will be assessed by visualizing the distribution of residuals using Q-Q plots to evaluate normality and scatterplots of predicted versus residual values to check for homoscedasticity. Model fit will be quantified using Marginal/Conditional $R^2$ values. In the event of significant violations of these assumptions, either appropriate data transformations (e.g., log-transformation) will be applied to the corresponding dependent variable or an alternative link function (e.g., logit link function) will be applied to the corresponding LME models to ensure the validity of the statistical inferences. Sensitivity analyses applicable in the current study, based on Thabane et al. [40] will include 1) assessment of outliers by z-score or boxplot, and subsequent analysis with/without outliers, 2) re-analyzing the dataset with only cases with complete testing across the timepoints, 3) perform analysis to assess the robustness of our findings across subject recruitment sites by including/

excluding the 'Recruitment Center' factor in the models, 4) analysis with/without adjustment for baseline characteristics (age, education, baseline CAB and baseline performance of the predicted variables), and 5) use of non-parametric methods (e.g., generalized linear models) vs. LME models to examine the robustness of the findings.

### Safety considerations

Exposure to VR is generally considered to have a low risk profile. Nevertheless, precautionary measures will be implemented to optimize the participant's safety and acceptability. In the screening session, each participant will have a 10-minute VR script training exposure in which s/he will experience visual/auditory input and engage in conversations/actions similar to those used in the therapy sessions. Those incompatible with the VR script training exposure will be excluded from the study. In each therapy session, PWA will receive a 5-minute break every 10-minute VR exposure. Therapy providers will monitor participants' conditions closely. A log sheet for reporting adverse events. The protocol for safety monitoring and adverse events reporting will follow the guidelines issued by the Institutional Review Board of the Hong Kong Polytechnic University, based on the National Institute on Aging Adverse Event and Serious Adverse Event Guidelines [41]. Any discomfort, such as cybersickness, headache, dizziness, or emotional distress induced by VR exposure, will be evaluated using the SSQ during the session, while the incidence will be reported to the principal investigator (WW) on the same day. If the weighted total severity score exceeds 10 in more than one session, indicating significant symptoms, the principal investigator (WW) will lead her research team member (HTK) in discussing study discontinuation with the PWA and their caregiver. PWA can withdraw from the study without any negative consequences.

### Reliability and treatment fidelity

Approximately 20% of the data, randomly sampled across arms, timepoints, and script types, will be blindly scored by the research postgraduate student and an independently trained speech therapist to examine intra- and inter-rater reliability using intraclass correlations. The acceptable reliability threshold will be 90% for both measures. If the reliability measures fall below the target, the principal investigator (WW) and co-author HTK will re-evaluate the scoring criteria and procedures with the research personnel involved in scoring and rescore the data concerned. Treatment fidelity will be monitored by the first author and an independent speech therapist with at least 3 years of experience in aphasia management, using an intervention fidelity checklist based on recommendations by the National Institute of Health Behavior Change Consortium [42]. Fidelity audits will be implemented every three months by the research team.

### Study limitations

The current study will recruit only Cantonese-speaking PWA; the generalizability of the study findings to speakers of other languages or Chinese dialects should be interpreted with caution. Additionally, during the screening process, PWA who cannot tolerate the 10-minute VR exposure will be excluded from the study. Such selection bias may affect the generalizability of the research findings to broader PWA populations.

### Timeline of the study

Participants will be recruited from 1st August 2025–31st December 2026. Assessment and treatment will span from September 2025 to May 2027. Data analysis and dissemination are scheduled to take place between June 2027 and December 2027.

### Data management plan

Data management will be overseen by the Principal Investigator (the first author) and co-authors. This process complies with the requirements of the Institutional Review Board of the Hong Kong Polytechnic University and is independent of

the funder. Physical records will be secured within locked file cabinets, with access strictly limited to the research team members. Digital materials, specifically audio recordings, will be stored on secure servers provided by the Hong Kong Polytechnic University and will be accessible only to the research team members. Any data shared on an open-access repository will first undergo de-identification using a unique subject code to preserve anonymity before distribution. Furthermore, any instance of unauthorized data access will be reported to the Institutional Review Board of the Hong Kong Polytechnic University.

## Dissemination plan

The research findings will be disseminated upon the conclusion of data collection and preliminary data analysis, estimated to occur within a four- to six- month timeframe. Initial results will be presented at local and overseas conferences on topics such as speech-language therapy, aphasia intervention and management, and the use of VR and other advanced technologies in rehabilitation. The Principal Investigator (WW) will assume leadership in preparing comprehensive reports and collaborating with co-authors to submit the findings to international peer-reviewed journals approximately six months after the final data analysis. Furthermore, face-to-face and/or online workshops introducing VR Script training, including its technical requirements, setup, and clinical efficacy, will be conducted for healthcare professionals and caregivers supporting aphasia rehabilitation in Hong Kong and neighboring cities. The anonymized dataset will be made accessible via an open-access data repository.

## Supporting information

**S1 File. SPIRIT Checklist.**
(PDF)

**S1 Fig. SPIRIT Schedule of enrollment, interventions, and assessments.**
(TIF)

**S1 Protocol. Protocol 1.0.**
(PDF)

## Acknowledgments

We are grateful to the following organizations for their support in participant recruitment in the project: AKA Social Service, The Hong Kong Polytechnic University Speech Therapy Unit, The Hong Kong Stroke Association, The Neighbourhood Advice-action Council NT West Community Rehabilitation Day Centre.

## Author contributions

**Conceptualization:** Winsy Wing Sze Wong, Donald Shi Pui Li.

**Data curation:** Winsy Wing Sze Wong, Donald Shi Pui Li, Kenneth Ngai Kuen Fong, Peter Hiu Fung Ng, Hoi Tsz Karen Kwok.

**Formal analysis:** Winsy Wing Sze Wong, Donald Shi Pui Li, Kenneth Ngai Kuen Fong, Peter Hiu Fung Ng, Hoi Tsz Karen Kwok.

**Funding acquisition:** Winsy Wing Sze Wong, Donald Shi Pui Li, Kenneth Ngai Kuen Fong, Peter Hiu Fung Ng.

**Investigation:** Winsy Wing Sze Wong.

**Methodology:** Winsy Wing Sze Wong, Donald Shi Pui Li, Kenneth Ngai Kuen Fong, Peter Hiu Fung Ng.

**Project administration:** Winsy Wing Sze Wong, Donald Shi Pui Li, Hoi Tsz Karen Kwok.

**Resources:** Winsy Wing Sze Wong, Donald Shi Pui Li, Hoi Tsz Karen Kwok.

**Writing – original draft:** Winsy Wing Sze Wong, Donald Shi Pui Li, Hoi Tsz Karen Kwok.

**Writing – review & editing:** Winsy Wing Sze Wong, Donald Shi Pui Li, Kenneth Ngai Kuen Fong, Peter Hiu Fung Ng, Hoi Tsz Karen Kwok.

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
