## [Decision Letter · Decision Letter 0]

10 Feb 2026

PONE-D-25-53607Can immersive virtual reality magnify treatment outcomes of computerized script training on Cantonese speakers with chronic aphasia? Protocol of a randomized controlled trialPLOS One

Dear Dr. Wong,

Thank you for submitting your manuscript to PLOS ONE. After careful consideration, we feel that it has merit but does not fully meet PLOS ONE’s publication criteria as it currently stands. Therefore, we invite you to submit a revised version of the manuscript that addresses the points raised during the review process.

Dear Authors,

Thank you for submitting your manuscript entitled

*“Can immersive virtual reality magnify treatment outcomes of computerized script training on Cantonese speakers with chronic aphasia? Protocol of a randomized controlled trial”* (PONE-D-25-53607) to PLOS ONE.

Your manuscript has now been reviewed by three independent reviewers. All reviewers agree that the study addresses an important and timely clinical question and that the overall randomized controlled design is appropriate. The trial is well motivated, registered, and supported by ethical approval, and the focus on Cantonese-speaking people with aphasia addresses an underrepresented population in the literature.

However, while Reviewers 1 and 2 raise primarily clarificatory and reporting-related issues, Reviewer 3 identifies several substantive methodological and procedural concerns that require resolution before the protocol can be considered suitable for publication. After careful consideration of all reviews, I have therefore decided that the manuscript requires Major Revision.

Below, I summarise the key issues that must be addressed in a revised version.

**Major issues requiring attention**

**Randomization and allocation concealment**
The protocol must provide a clear and explicit description of allocation concealment procedures, including who generates the randomization sequence, who enrolls participants, and who assigns interventions. Please also clarify whether stratified or minimization procedures will be used to balance key prognostic variables across the three arms, or justify why this is not planned.
**Primary outcome definition and scoring procedures**
The primary outcome requires more precise operationalization. Please clarify criteria for correct script production, handling of partial responses, self-corrections, and semantic substitutions.In addition, scoring procedures must be specified in greater detail, including the number of raters, blinding procedures, rater training, adjudication rules, and justification or validation of the adapted scoring approach (e.g., NORLA-6 adaptation).
**Reliability and double scoring**
The current plan for double scoring a limited proportion of recordings requires strengthening. Please increase and pre-specify the proportion of recordings to be double scored, define acceptable reliability thresholds, and outline procedures if reliability falls below target.
**Sample size calculation and assumptions**
Please provide a fully transparent account of the sample size calculation, including justification for the assumed effect size, statistical test framework (e.g., LME vs. repeated-measures ANOVA), assumptions regarding correlations among repeated measures, and how power relates to specific planned comparisons. Attrition assumptions should also be clarified.
**Statistical analysis plan (pre-specification)**
The statistical analysis plan requires further pre-specification, including the planned mixed-effects model structure, handling of baseline measures and covariates, treatment of missing data, sensitivity analyses, and approach to multiple comparisons. Planned subgroup or moderator analyses should be clearly identified as confirmatory or exploratory.
**Blinding and bias mitigation**
Please clarify how assessor blinding will be maintained and monitored, and how potential expectancy effects related to unblinded treatment providers will be mitigated.
**Treatment fidelity and therapist training**
Additional details are required regarding therapist training, treatment manuals, fidelity monitoring procedures, and corrective actions for protocol deviations.
**Safety monitoring and adverse events**
The protocol should specify procedures for monitoring and reporting adverse events (including VR-related symptoms), criteria for discontinuation, and the structure of safety oversight (e.g., DSMB or equivalent justification).
**Eligibility criteria and generalizability**
Please clarify inclusion and exclusion criteria (including aphasia severity and motor speech disorders) and discuss how VR tolerance-based exclusions may affect generalizability.

**Additional comments**

In addition to the major points above, please address the minor and editorial comments from all reviewers, including clarifying timelines, outcome measurement details, data-sharing plans, adherence to reporting guidelines (e.g., SPIRIT), and language clarity.

**Revision instructions**

Please submit a detailed, point-by-point response to all reviewer comments, clearly indicating how each issue has been addressed. Revised text should be clearly identifiable.

Please note that failure to adequately address the major methodological and procedural concerns outlined above may preclude further consideration of the manuscript.

We appreciate the potential contribution of this protocol and look forward to receiving a carefully revised version.

Sincerely,

Jan Christopher Cwik

We look forward to receiving your revised manuscript.

Kind regards,

Jan Christopher Cwik, Prof. Dr. Dr.

Academic Editor

PLOS One

**Journal Requirements:**

1. When submitting your revision, we need you to address these additional requirements. Please ensure that your manuscript meets PLOS ONE's style requirements, including those for file naming. The PLOS ONE style templates can be found at https://journals.plos.org/plosone/s/file?id=wjVg/PLOSOne_formatting_sample_main_body.pdf and https://journals.plos.org/plosone/s/file?id=ba62/PLOSOne_formatting_sample_title_authors_affiliations.pdf 2. We note that the grant information you provided in the ‘Funding Information’ and ‘Financial Disclosure’ sections do not match.  When you resubmit, please ensure that you provide the correct grant numbers for the awards you received for your study in the ‘Funding Information’ section. 3. Thank you for stating the following in the Acknowledgments Section of your manuscript: We are grateful to the following organizations for their support in subject recruitment in the funding application: AKA Social Service, The Hong Kong Polytechnic University Speech Therapy Unit, The Hong Kong Stroke Association, The Neighbourhood Advice-action Council NT West Community Rehabilitation Day Centre. We note that you have provided funding information that is not currently declared in your Funding Statement. However, funding information should not appear in the Acknowledgments section or other areas of your manuscript. We will only publish funding information present in the Funding Statement section of the online submission form. Please remove any funding-related text from the manuscript and let us know how you would like to update your Funding Statement. Currently, your Funding Statement reads as follows: Wong received the General Research Fund from Hong Kong Research Grants Council for the project (Project number: 15602024).https://www.ugc.edu.hk/eng/rgc/funding_opport/grf/The funders had no role in study design, data collection and analysis, decision to publish, or preparation of the manuscript  Please include your amended statements within your cover letter; we will change the online submission form on your behalf. 4. Please note that your Data Availability Statement is currently missing the repository name and/or the DOI/accession number of each dataset and a direct link to access each database. If your manuscript is accepted for publication, you will be asked to provide these details on a very short timeline. We therefore suggest that you provide this information now, though we will not hold up the peer review process if you are unable. 5. Your ethics statement should only appear in the Methods section of your manuscript. If your ethics statement is written in any section besides the Methods, please move it to the Methods section and delete it from any other section. Please ensure that your ethics statement is included in your manuscript, as the ethics statement entered into the online submission form will not be published alongside your manuscript. 6. If the reviewer comments include a recommendation to cite specific previously published works, please review and evaluate these publications to determine whether they are relevant and should be cited. There is no requirement to cite these works unless the editor has indicated otherwise.

Reviewers' comments:

Reviewer's Responses to Questions

**Comments to the Author**

1. Does the manuscript provide a valid rationale for the proposed study, with clearly identified and justified research questions?

Reviewer #1: Yes

Reviewer #2: Yes

Reviewer #3: Yes

2. Is the protocol technically sound and planned in a manner that will lead to a meaningful outcome and allow testing the stated hypotheses?

Reviewer #1: Yes

Reviewer #2: Partly

Reviewer #3: Partly

3. Is the methodology feasible and described in sufficient detail to allow the work to be replicable?

Reviewer #1: Yes

Reviewer #2: Yes

Reviewer #3: Yes

4. Have the authors described where all data underlying the findings will be made available when the study is complete?

Reviewer #1: Yes

Reviewer #2: Yes

Reviewer #3: Yes

5. Is the manuscript presented in an intelligible fashion and written in standard English?

Reviewer #1: Yes

Reviewer #2: No

Reviewer #3: Yes

6. Review Comments to the Author

You may also provide optional suggestions and comments to authors that they might find helpful in planning their study.

**Reviewer #1:** For sample size calculation, what test was used, t-test? Was a repeated measure method considered?

Descriptive statistics can be added to the statistical analysis plan.

Was baseline considered as a covariate in the LME? Change from baseline can be used as the dependent variable.

P value adjustment needs to be clearly written.

**Reviewer #2:** The study is well-grounded, and the authors demonstrate a solid understanding of the subject matter and relevant literature. The work addresses an important need to strengthen the validity of studies conducted using virtual reality (VR).

However, several points warrant reconsideration:

Protocol Design and Covariates: The study design does not account for covariates such as age, severity, and time since stroke, and the power calculation was performed without considering these factors. Additionally, these covariates are not planned to be balanced between groups. This represents a significant limitation that could reduce the validity of the results.

Population Specificity: The study focuses on Cantonese-speaking participants. While this is appropriate from a study design perspective, it should be clearly acknowledged as a limitation in the manuscript.

Subjective Assessments: Some subjective evaluations are conducted by a single author. It may be advisable for multiple evaluators to perform these assessments independently, followed by a joint review of scores, to improve reliability.

General Writing and Clarity: The manuscript has issues with overall fluency, and certain sections contain grammatical errors. A thorough revision of the text is recommended to improve readability and clarity.

Overall, the study has strong potential, but addressing the points above would enhance its rigor and the clarity of its presentation.

**Reviewer #3:** - Well-motivated, timely three-arm RCT (VR vs non-VR computerized script training vs no-treatment) targeting an important clinical question for Cantonese-speaking PWA. Trial is registered (NCT06722092), ethics approved (PolyU IRB), and data/code sharing on OSF is declared. The basic design and choice of script training are appropriate; however, several methodological, safety, and analysis details require clarification or strengthening before the protocol is acceptable.

Major issues (expanded)

1. Randomization and allocation concealment

- Provide explicit allocation concealment procedures: who generates and stores the sequence, who enrolls participants, and who performs assignments. Describe physical/technical safeguards (e.g., opaque sealed envelopes, central web-based randomization).

- Consider stratified randomization or minimization on key prognostic variables (baseline AQ/aphasia severity, time since stroke, age) to prevent imbalance across three arms.

2. Primary outcome definition, scoring, and rater procedures

- Fully operationalize the primary outcome: precise criteria for “correct” script production (treatment of semantic substitutions, paraphasias, partial utterances, prompts), timing window, and handling of self-corrections.

- Provide details on scoring procedures: blinded independent raters, number of raters per recording, training, and adjudication rules.

- The NORLA-6 adaptation and counting “semantically acceptable” productions as correct require justification or validation evidence (pilot reliability/validity). If novel, include plan to validate scoring (inter-rater reliability targets, pilot ICCs).

3. Reliability and proportion of double scoring

- Ten percent double-scoring is low. Increase to at least 20–25% of recordings, sampled across arms, timepoints, and script types. Pre-specify acceptable ICC thresholds and procedures if reliability is below threshold (retraining, rescoring).

4. Sample size calculation and assumptions

- Report full sample-size inputs: effect size origin and justification for d=0.74, assumed correlation among repeated measures, number of repeated measures, alpha adjustment for multiple pairwise comparisons (if applicable), and statistical test basis (LME vs repeated measures ANOVA). Clarify whether power is for VR vs non-VR, VR vs control, or all comparisons.

- Explain how the 20% attrition was applied and whether recruitment targets account for missing-at-random vs differential dropout across arms.

5. Statistical analysis plan (pre-specify)

- Provide detailed LME specification: random intercepts and/or slopes, whether time is categorical or continuous, covariance structure, fixed effects (arm, time, arm*time), planned covariates (age, baseline AQ, education) and rationale.

- State handling of missing data (ML via LME, multiple imputation), sensitivity analyses, and assumptions to be tested.

- Predefine primary contrasts and multiple comparison correction approach (hierarchical testing vs Bonferroni). Specify planned subgroup/moderator analyses (aphasia type, severity) and whether they are exploratory.

6. Blinding and bias mitigation

- Describe concrete steps to maintain assessor blinding (separate staff, standardized scripts to avoid revealing arm, blinding checks), and plan to assess and report blinding success.

- Clarify that therapy deliverers cannot be blinded and how expectancy effects will be mitigated (standardized instructions, scripted feedback).

7. Treatment fidelity and therapist training

- Provide details on manuals, training sessions for therapists/assistants, fidelity checklists, frequency of fidelity audits, and predefined corrective actions if deviations occur. Specify who monitors fidelity (independent reviewer) and their criteria.

8. Safety monitoring and adverse events

- Specify adverse-event monitoring procedures: which events are tracked (cybersickness, headache, dizziness, emotional distress), how often assessed, reporting timelines, criteria for temporary/permanent discontinuation, and who adjudicates events.

- State whether a DSMB or safety oversight committee will be convened; if not, justify and outline alternative safety oversight.

9. Inclusion/exclusion and generalizability

- Excluding anyone intolerant to a 10-minute VR exposure introduces selection bias toward those tolerating VR; discuss implications for generalizability and how findings will be interpreted relative to broader PWA populations.

- Clarify lower AQ limit (severe aphasia allowed?) and operational criteria for excluding “moderate to severe” motor speech disorders.

Minor issues and reporting suggestions

1. Sample and timeline consistency

- Reconcile inconsistent recruitment/start dates (May vs Aug 15, 2025) and provide final clear timeline. Provide updated, high-resolution SPIRIT figure

2. Outcome measurement details

- For CANELT z-scores and other transformed measures, specify normative data sources and transformation procedures.

- Clarify use of averaged baseline (two pre-treatment assessments) vs single baseline value in analyses.

3. Data sharing and reproducibility

- Good to see OSF links. Specify exactly which materials will be shared (de-identified audio, transcripts, scoring sheets, analysis code), timing of release, and de-identification steps for audio files.

4. Reporting and adherence to guidelines

- Ensure completed SPIRIT checklist and adherence to CONSORT extension for non-pharmacologic trials and ROMA recommendations are demonstrated in the final protocol.

5. Figure/table quality and minor edits

- Improve figure resolution and correct small typographical repetitions in submission metadata.

7. PLOS authors have the option to publish the peer review history of their article (what does this mean?). If published, this will include your full peer review and any attached files.

Reviewer #1: No

Reviewer #2: No

Reviewer #3: **Yes:**lars clemmensen

You may also use PLOS’s free figure tool, NAAS, to help you prepare publication quality figures: https://journals.plos.org/plosone/s/figures#loc-tools-for-figure-preparation

---

## [Author Response · Author response to Decision Letter 1]

28 Apr 2026

Dear Editor,

Response letter to the submission “Can immersive virtual reality magnify treatment outcomes of computerized script training on Cantonese speakers with chronic aphasia? Protocol of a randomized controlled trial” (PONE-D-25-53607 & PONE-D-25-53607R1)

Thank you very much to the reviewers and editor for their comments and suggestions. We have considered them seriously and addressed them appropriately. Please find our point-by-point responses below.

Should you have any questions, please feel free to let us know.

Best,

Winsy Wing-Sze Wong (Corresponding author)

PONE-D-25-53607R1

Please upload a Response to Reviewers letter which should include a point by point response to each of the points made by the Editor and / or Reviewers. (This should be uploaded as a 'Response to Reviewers' file type.)

Response: The Response to Reviewers letter is uploaded to the system.

Please upload a copy of Figure 2

Response: The supporting information has been modified in the manuscript. In accordance with PLOS ONE editorial guidelines, the SPIRIT checklist (PDF) and SPIRIT Schedule (TIFF) have been uploaded as S1 Supporting Document and S1 Figure, respectively. Additionally, Protocol 1.0 has been uploaded as S1 Protocol in supporting information. Please note that Figure 2 is not included in the submission. Statement of fig2a is removed from line 303-304.

PONE-D-25-53607

Major issues requiring attention

1. Randomization and allocation concealment

The protocol must provide a clear and explicit description of allocation concealment procedures, including who generates the randomization sequence, who enrolls participants, and who assigns interventions. Please also clarify whether stratified or minimization procedures will be used to balance key prognostic variables across the three arms, or justify why this is not planned.

Response: An explicit description of randomization and allocation concealment had been added under the Participant section (lines 187-204) and Randomization (lines 219-228). We did not plan to balance the prognostic variables (e.g., age, aphasia severity, and lesion size) in the condition assignment due to practicality issues. For instance, most of the stroke patients in Hong Kong did not have MRI scans post-stroke due to resource limitations. It is therefore impossible to consider all these variables during condition allocation. We fully understand their potential impacts on aphasia therapy outcomes; these variables (where available) will be included as covariates in the analysis, illustrated in the LME models (lines 385-443).

2. Primary outcome definition and scoring procedures

The primary outcome requires more precise operationalization. Please clarify criteria for correct script production, handling of partial responses, self-corrections, and semantic substitutions.

In addition, scoring procedures must be specified in greater detail, including the number of raters, blinding procedures, rater training, adjudication rules, and justification or validation of the adapted scoring approach (e.g., NORLA-6 adaptation).

Response: Scoring will be performed by the fifth author, trained speech-language pathology students, and trained personnel with backgrounds in speech-language pathology or linguistics (lines 345-349). Assessors will be blinded to the intervention condition. Patients will be reminded to keep their intervention condition confidential during assessment. Blinding procedures for assessment and treatment are detailed in lines 266-271.

Adaptation of NORLA-6 concerns the deletion of rules that are not applicable to Cantonese (including prosodic variation of rating 4 and verb tense use in rating 3; please refer to Table 3 for details). The remaining scoring criteria will be consistent with the NORLA-6 framework. Detailed scoring procedures and criteria are mentioned in lines 345-371.

3. Reliability and double scoring

The current plan for double scoring a limited proportion of recordings requires strengthening. Please increase and pre-specify the proportion of recordings to be double scored, define acceptable reliability thresholds, and outline procedures if reliability falls below target.

Response: The proportion of recordings to be double-scored has been increased to 20%. The reliability threshold and contingency procedures for low-reliability are detailed in lines 468-471.

4. Sample size calculation and assumptions

Please provide a fully transparent account of the sample size calculation, including justification for the assumed effect size, statistical test framework (e.g., LME vs. repeated-measures ANOVA), assumptions regarding correlations among repeated measures, and how power relates to specific planned comparisons. Attrition assumptions should also be clarified.

Response: Sample size estimation was conducted using G*Power for repeated measurements, within-between interaction ANOVA. Lines 187-204 detailed the sample size calculations and assumptions, including a clarified rationale for the expected attrition rate.

5. Statistical analysis plan (pre-specification)

The statistical analysis plan requires further pre-specification, including the planned mixed-effects model structure, handling of baseline measures and covariates, treatment of missing data, sensitivity analyses, and approach to multiple comparisons. Planned subgroup or moderator analyses should be clearly identified as confirmatory or exploratory.

Response: Further details regarding the pre-specification of the statistical analysis, including the planned mixed-effects model structure, handling of baseline measures and covariates, treatment of missing data, sensitivity analyses, and approach to multiple comparisons, have been provided in lines 371-443. Subgroup analyses are exploratory (lines 426-427).

6. Blinding and bias mitigation

Please clarify how assessor blinding will be maintained and monitored, and how potential expectancy effects related to unblinded treatment providers will be mitigated.

Response: As assessors will be blinded to the intervention condition, patients will be reminded to keep their intervention condition confidential during assessment. Blinding procedures for assessment and treatment are detailed in lines 266-271. Further details regarding bias mitigation have been addressed in lines 281-294.

7. Treatment fidelity and therapist training

Additional details are required regarding therapist training, treatment manuals, fidelity monitoring procedures, and corrective actions for protocol deviations.

Response: Therapist training and treatment manuals will be monitored and developed by the authors (WW and HTK). Fidelity monitoring will be conducted by the author (WW) and an independent speech therapist, while the PI and the research team will oversee corrective actions for protocol deviations. These procedures have been addressed in lines 281-294 and lines 471-475.

8. Safety monitoring and adverse events

The protocol should specify procedures for monitoring and reporting adverse events (including VR-related symptoms), criteria for discontinuation, and the structure of safety oversight (e.g., DSMB or equivalent justification).

Response: The project will follow safety guidelines issued by the Institutional Review Board of the Hong Kong Polytechnic University, based on the National Institute on Aging Adverse Event and Serious Adverse Event Guidelines lines 451-457. Specific procedures for adverse event monitoring have been further addressed in lines 457-463.

9. Eligibility criteria and generalizability

Please clarify inclusion and exclusion criteria (including aphasia severity and motor speech disorders) and discuss how VR tolerance-based exclusions may affect generalizability.

Response: Criteria for defining the severity of aphasia and motor speech disorders are clarified (lines 205-207, 209-210). The study includes PWA with an aphasia quotient below 96.4 and no motor speech disorders of moderate to severe level according to the Therapy Outcome Measurement. The exclusion of PWA unable to tolerate 10 minutes of VR exposure may result in selection bias, which might impact the generalizability of the results. Detailed clarification of the inclusion and exclusion criteria can be found in lines 205-217. We agree that VR-based exclusions may affect generalizability, which has been included as a limitation of the study (lines 479-481).

Additional comments

In addition to the major points above, please address the minor and editorial comments from all reviewers, including clarifying timelines, outcome measurement details, data-sharing plans, adherence to reporting guidelines (e.g., SPIRIT), and language clarity.

Response: The above-mentioned comments have been addressed in response to the reviewer’s feedback (please refer to the specific responses below).

Journal Requirements:

Response: The manuscript has been edited according to PLOS ONE’s style requirements.

Response: The name of the funder has been corrected in the ‘Funding Information’ section.

We are grateful to the following organizations for their support in subject recruitment in the funding application: AKA Social Service, The Hong Kong Polytechnic University Speech Therapy Unit, The Hong Kong Stroke Association, The Neighbourhood Advice-action Council NT West Community Rehabilitation Day Centre.

Wong received the General Research Fund from Hong Kong Research Grants Council for the project (Project number: 15602024).

https://www.ugc.edu.hk/eng/rgc/funding_opport/grf/

The funders had no role in study design, data collection and analysis, decision to publish, or preparation of the manuscript

Response: Regarding the Acknowledgments section, we clarify that the mentioned organizations provided support solely for participant recruitment and were not involved in funding acquisition; amendments have been made to lines 511-514.

4. Please note that your Data Availability Statement is currently missing the repository name and/or the DOI/accession number of each dataset and a direct link to access each database. If your manuscript is accepted for publication, you will be asked to provide these details on a very short timeline. We therefore suggest that you provide this information now, though we will not hold up the peer review process if you are unable.

Response: The links for the code and data have been added to the Data availability (lines 520-522) and Code availability sections (lines 525-526).

Response: The ethics statement has been relocated to the Methods section in lines 181-185.

Response: Not applicable.

Reviewers' comments:

Reviewer's Responses to Questions

Reviewer #1: For sample size calculation, what test was used, t-test? Was a repeated measure method considered?

Descriptive statistics can be added to the statistical analysis plan.

Was baseline considered as a covariate in the LME? Change from baseline can be used as the dependent variable.

P value adjustment needs to be clearly written.

Response: Sample size estimation was conducted using G*Power (version 3.1.9.7) for repeated measurements, within-between interaction ANOVA. The detailed calculation methodology is provided in lines 187-204. Descriptive statistics for outcome measures and characteristics of PWA have been added under the section of Data Analysis (lines 371-372). We agree that baseline performance should be included in the LME and each model is specified in lines 372-419. P-value adjustment has been clarified (lines 425-426).

Reviewer #2: The study is well-grounded, and the authors demonstrate a solid understanding of the subject matter and relevant literature. The work addresses an important need to strengthen the validity of studies conducted using virtual reality (VR).

However, several points warrant reconsideration:

Protocol Design and Covariates: The study design does not account for covariates such as age, severity, and time since stroke, and the power calculation was performed without considering these factors. Additionally, these covariates are not planned to be balanced between groups. This represents a significant limitation that could reduce the validity of the results.

Response: Due to practicality issues, we did not plan to balance the factors in condition allocation. We did not plan to balance the prognostic variables (e.g., age, aphasia severity, and lesion size) in the condition assignment due to practicality issues. For instance, most of the stroke patients in Hong Kong did not have MRI scans post-stroke due to resource limitations. It is therefore impossible to consider all these variables during condition allocation. We fully understand their potential impacts on aphasia therapy outcomes; these variables (where available) will be included as covariates in the analysis, illustrated in the LME models (lines:385-419).

Population Specificity: The study focuses on Cantonese-speaking participants. While this is appropriate from a study design perspective, it should be clearly acknowledged as a limitation in the manuscript.

Response: We thank the reviewer for raising this. Population specificity has been addressed in the limitation section (lines 477-478).

Subjective Assessments: Some subjective evaluations are conducted by a single author. It may be advisable for multiple evaluators to perform these assessments independently, followed by a joint review of scores, to improve reliability.

Response: We agree with the reviewer’s suggestions. Scoring and subjective evaluation will be conducted by the fifth author (HTK), trained speech language pathology students and trained personnel with speech language pathology or linguistics background (lines 345-349).

General Writing and Clarity: The manuscript has issues with overall fluency, and certain sections contain grammatical errors. A thorough revision of the text is recommended to improve readability and clarity.

Response: The manuscript has been proofread by a professional copywriter for editorial clarity.

Overall, the study has strong potential, but addressing the points above

---

## [Decision Letter · Decision Letter 1]

13 May 2026

Can immersive virtual reality magnify treatment outcomes of computerized script training on Cantonese speakers with chronic aphasia? Protocol of a randomized controlled trial

PONE-D-25-53607R1

Dear Dr. Wong,

We’re pleased to inform you that your manuscript has been judged scientifically suitable for publication and will be formally accepted for publication once it meets all outstanding technical requirements.

Kind regards,

Jan Christopher Cwik, Prof. Dr. Dr.

Academic Editor

PLOS One

Reviewers' comments:

Reviewer's Responses to Questions

**Comments to the Author**

1. Does the manuscript provide a valid rationale for the proposed study, with clearly identified and justified research questions?

Reviewer #1: Yes

2. Is the protocol technically sound and planned in a manner that will lead to a meaningful outcome and allow testing the stated hypotheses?

Reviewer #1: Yes

3. Is the methodology feasible and described in sufficient detail to allow the work to be replicable?

Reviewer #1: Yes

4. Have the authors described where all data underlying the findings will be made available when the study is complete?

Reviewer #1: Yes

5. Is the manuscript presented in an intelligible fashion and written in standard English?

Reviewer #1: Yes

6. Review Comments to the Author

You may also provide optional suggestions and comments to authors that they might find helpful in planning their study.

Reviewer #1: All my concerns are addressed.

7. PLOS authors have the option to publish the peer review history of their article (what does this mean?). If published, this will include your full peer review and any attached files.

Reviewer #1: No

---

## [Editor Report · Acceptance letter]

PONE-D-25-53607R1

PLOS One

Dear Dr. Wong,

I'm pleased to inform you that your manuscript has been deemed suitable for publication in PLOS One. Congratulations! Your manuscript is now being handed over to our production team.

Kind regards,

on behalf of

Prof. Dr. Dr. Jan Christopher Cwik

Academic Editor

PLOS One